# Efficacy of Pectin-Based Coating Added with a Lemon Byproduct Extract on Quality Preservation of Fresh-Cut Carrots

**DOI:** 10.3390/foods11091314

**Published:** 2022-04-30

**Authors:** Valeria Imeneo, Amalia Piscopo, Olga Martín-Belloso, Robert Soliva-Fortuny

**Affiliations:** 1Department of Food, Environmental and Nutritional Sciences (DeFENS), University of Milan, G. Celoria 2, 20133 Milan, Italy; valeria.imeneo@unimi.it; 2Department of AGRARIA, University Mediterranea of Reggio Calabria, Vito, 89124 Reggio Calabria, Italy; amalia.piscopo@unirc.it; 3Department of Food Technology, University of Lleida—Agrotecnio CERCA Center, Av. Alcalde Rovira Roure 191, 25198 Lleida, Spain; olga.martin@udl.cat

**Keywords:** pectin-based coating, quality attributes, fresh-cut carrot, respiratory activity, carotenoids and phenolic compounds, lemon byproduct, antioxidant

## Abstract

The effect of an edible pectin-based coating supplemented with a lemon byproduct extract on the quality attributes of fresh-cut carrots was studied. Color, hardness, microbial growth, respiratory activity, and antioxidant properties of fresh-cut carrots were studied during 14 days of storage at 4 °C. The application of a pectin-based coating containing a lemon byproduct extract preserved carrots’ physiological parameters, reduced their physiological activity and, thus, delayed senescence. This aspect was also confirmed by the reduced O_2_ consumption of the coated carrots due to the slowing down of the product’s metabolic reactions. Moreover, coated carrots were characterized by limited changes in colour (ΔE < 3) and white-blush development on both cortical tissue and vascular cylinder, and the presence of calcium chloride in the coating formulation helped to maintain carrots’ hardness throughout storage. In addition, treatment with pectin-based coating and lemon byproduct extract improved microbiological stability of fresh-cut carrots, showing the lowest value of total bacterial count immediately after treatment (2.58 log CFU g^−1^). This kind of treatment also resulted in a significant preservation of valuable compounds (17.22 mg GAE 100 g^−1^) and antioxidant activity level (289.49 µM Trolox 100 g^−1^), reducing the wounding stress induced by processing operations for at least ten days.

## 1. Introduction

Consumption of fresh fruit and vegetables is known to be beneficial to human health. In this context, carrot consumption is becoming ever more popular thanks to their nutritional value. Carrots represent a good source of bioactive compounds, namely, fibre and antioxidants such as carotenoids, phenolic compounds, and vitamin C [1,2]. However, like other fresh fruit and vegetables, they are highly perishable. During post-harvest handling and storage, significant losses of vitamins and other phytonutrients can occur, depending on the nutrient, genotype, physical damage, temperature and storage conditions [3]. The colour and appearance of minimally processed carrots are critical quality attributes. Their characteristic bright orange colour may be rapidly lost due to dehydration and the development of white blush on the surface, hence reducing their acceptability [4,5]. Maintaining the overall quality of minimally processed fruit and vegetables becomes even more difficult. Therefore, fresh-cut fruit and vegetables have become a challenging problem among food scientists and technologists, over the years [6].

In this respect, the application of techniques such as the dipping of minimally processed carrots in acid solutions demonstrated to be an efficient approach to preserve their quality, limit colour changing and microbial growth, reduce enzymatic activity and increase the preservation of total carotenoid content [5].

Moreover, polysaccharide edible coatings have proven to be a valid strategy to boost food appearance and extend the shelf-life of fresh-cut fruits and vegetables. Coatings can form a semi-permeable barrier to gasses and be potential carriers of additives to help in preserving or improving the quality of produce [6]. Polysaccharide-based edible coatings can help to prolong the shelf life of fresh fruits and vegetables by slowing down decay processes associated to water loss [7]. Among polysaccharides, pectins are valuable compounds for coating formulations thanks to their ability to create rigid and stable gels, enabling effective food applications [8,9,10,11]. Pectins are α-1,4 bonded D-galacturonic acid polymers characterized by the presence of hydrophobic groups, consisting of methyl ester, acetyl, and protein residues, that promote the absorption of organic lipid substances, hence contributing to their emulsifying capacity [12]. In particular, the hydrocolloidal and polyelectrolyte properties of pectins define their unique capacities, such as high water retention in colloidal systems and their stabilization, and aptitude to plasticize with glycerol [13]. According to their content of methyl esters or the degree of esterification (DE), pectins are divided into high methoxyl (DE > 50%) or low methoxyl (DE < 50%), showing a significant influence on solubility and gel-formation characteristics [14]. In coating formulations, low-methoxyl pectins are generally used, mainly because of their ability to develop strong gels or insoluble polymers when multivalent metal cations, such as calcium, are present [15,16,17]. Overall, edible coatings are applied on fruit and vegetables due to their ability to reduce respiration and senescence, retain moisture and slow down colour changes throughout storage [6,18]. Several authors have studied the incorporation of additives, namely, antioxidant, antimicrobial and antibrowning compounds or texture enhancers, into edible coating formulations, making them a valid option for decreasing physiological postharvest deterioration rate in minimally processed fruits and vegetables [19,20,21,22]. The application of plant byproducts and extracts is regarded as a way of incorporating these active compounds to fresh-cut fruit and vegetables while maintaining a label free from synthetic additives. In this regard, lemon byproducts, consisting of lemon peel, pulp and seeds, represent a natural source of minerals, organic acids, dietary fibre and phenolic compounds, such as phenolic acids and flavonoids (flavanones, flavonols, flavones), characterised by antioxidant, anti-inflammatory and antimicrobial properties [23,24,25,26,27,28]. Among these bioactive compounds, flavanones and flavones are the most plentiful flavonoids, followed by neohesperidin, naringin, rutin and apigenin. Other compounds identified in lemon peels are furocoumarins and coumarins. The highest quantity of phenolic acids is found in lemon seeds, the only waste characterised by the presence of gallic acid, protocatechic acid and p-cumaric acid, as well as obacunone, a compound belonging to the class of limonoids [29]. In view of the antioxidant and antimicrobial activities exhibited by these compounds, the goal of this study was to determine the effect of an edible pectin-based coating supplemented with a lemon byproduct extract on the quality attributes of fresh-cut carrots. This approach would contribute to the shelf-life extension of the ready-to-eat commodity, while valorising the potential of a byproduct generated from the citrus industry.

## 2. Materials and Methods

### 2.1. Chemical and Reagents

Food-grade low-methoxyl pectin (~30% esterified) (Sigma–Aldrich Chemic, Steinhein, Germany) was the carbohydrate biopolymer used to prepare the coating formulations. Glycerol (Merck, Whitehouse Station, NJ, USA) was used as plasticizer. Calcium chloride (Sigma–Aldrich Chemic, Steinhein, Germany) was added to promote pectin gelation by crosslinking. Folin-Ciocalteu reagent and ethanol were purchased from Scharlau S.L. (Barcelona, Spain); sodium carbonate was purchased from Fisher Scientific Scharlau Chemie (Loughborough, UK). DPPH (2,2-diphenyl-1-picrylhydrazyl), ABTS (2,2’-azino-bis acid (3- ethylbenzothiazolin-6-sulfonic acid), gallic acid and Trolox (6-hydroxy-2,5,7,8-tetramethylchroman-2-carboxylic acid) were bought from Sigma-Aldrich (St. Louis, MO, USA). Sodium hydroxide was acquired from Acros Organics (New Jersey, NJ, USA). Methanol was acquired from J.T. Baker S.A. (Sowińskiego, Poland). Acetone was purchased from Fisher Chemical (Loughborough, UK). Phenolphthalein was bought from POCH S.A. (Sowińskiego, Poland). PCA, GCA and buffer peptone water were purchased from Biokar Diagnostics (Beauvais, France). Sodium hypochlorite was purchased from Productes Sant Mateu (Barcelona, Spain).

### 2.2. Lemon Extract (LE)

Lemon byproducts (*Citrus limon (L.) Osbeck)*, consisting of peels, pulp and seeds, obtained after the extraction of lemon juice and essential oils, were provided by Agrumaria Reggina company, situated in Reggio Calabria (Italy). The lemon byproduct extract (LE) used in this study was obtained by an aqueous extraction at 70 °C for 30 min, as reported and characterised by Imeneo et al. [30].

### 2.3. Preparation of the Dipping and Coating Solutions

Carrots (*Daucus carota* cv. Nantes) were purchased in a local supermarket in Lleida (Spain). The carrots were carried to the laboratory of the University of Lleida and immediately processed for this study. Whole carrots (length of 17 ± 3 cm) were sanitized in a 200 µL L^−1^ NaClO solution for 2 min. Sanitized carrots were rinsed with tap water and the excess of water was blotted away at room temperature. Carrots were peeled and afterwards cut lengthwise into two parts and then transversely in two semi-discs (diameter 28 ± 2, height 20 ± 2 mm).

A portion of the carrot pieces (W + LE) was dipped in an aqueous solution containing LE (1%, *v*/*v*). Other carrot pieces were coated using a pectin-based coating (PC), prepared in accordance with Oms-Oliu et al. [15] description. The coating solution was prepared by dissolving pectin (2%, *w*/*v*) powder in distilled water and heating at 70 °C while stirring, until the solution became clear. Glycerol (1.5%, *w*/*v*) was added as a plasticizer to the pectin solution and the carrot pieces were dipped into it for 2 min. Excess coating material was drained for 1 min. Subsequently, carrot pieces were dipped for 2 min into a calcium-chloride aqueous solution (2%, *w*/*v*) with the addition of LE (1%, *v*/*v*) for crosslinking pectin (PC + LE).

Treatments without the addition of LE (W and PC) were prepared as control references.

Then, 100 g of carrots were placed in manually closed, side-perforated polyethylene terephthalate trays (150 mL) and stored at 4 ± 1 °C in the dark until they were withdrawn for analyses after 1, 3, 7, 10 and 14 days of storage.

Two trays of each treatment condition were taken at each monitoring time to carry out repetition of analyses.

### 2.4. Physicochemical Properties

Titratable acidity (% of citric acid) and pH (pH meter Crison micropH 2000, Crison Instruments S.A., Alella, Barcelona, Spain) were evaluated according to AOAC methods [31,32].

Total soluble solids (°Brix) were determined through the measurement of the refraction index with a digital refractometer (PR-32, 3412-J01, Atago Company Ltd., Tokyo, Japan) at 25 °C.

Dry matter (%) of carrot pieces was quantified according to AOAC method [33], by calculating loss weight at 70 °C until constant values were reached.

Colour

Colour values of cortical tissue (external side) and vascular cylinder (internal side) were measured with a colorimeter (Minolta Chroma Meter Model CR-400, Minolta Sensing Inc., Osaka, Japan). Ten readings were performed on each replicate. Total colour difference (ΔE), as reported by Thompson [34], and the whiteness index (WI), according to Piscopo et al. [5], were determined, considering CIELab coordinates in freshly cut carrot pieces and the values obtained at each day of monitoring according to the equations below: (1)$$\Delta E=\sqrt{\left[{\left(L-L_{0}\right)}^{2}+{\left(a-a_{0}\right)}^{2}+{\left(b-b_{0}\right)}^{2}\right]}$$
(2)$$\mathrm{WI}=100-\sqrt{\left[\left(100-L^{2}\right)+{\;a}^{2}+b^{2}\right]}$$
where *L*_0_, *a*_0_, and *b*_0_ are the values measured on the 1st day, and *L*, *a* and *b* relate to data measured at each sampling time.

Hardness

Hardness of cortical tissue and vascular cylinder of carrots was measured with a TA-XT2 texture analyser (Stable Micro Systems Ltd., Surrey, UK), fitted with a 4-mm-diameter cylinder steel probe, which went through the carrot surface for 10 mm at a constant speed of 5 mm·s^−1^ and automatic return. The probe movement was orthogonal to the carrot tissue. Hardness (N s^−1^) was defined as the area under the curve between the graphic depicting force vs time [35]. Five randomly withdrawn carrot pieces were analysed per treatment and sampling time.

### 2.5. Microbiological Analysis

Total aerobic bacterial counts (TBC) were evaluated throughout storage. Two replicate packages were analysed per treatment and sampling time and two counts were obtained from each one. Under sterile conditions, 10 g of carrot pieces were homogenized for 3 min with 90 mL of 0.1% sterile peptone water using a Stomacher Lab Blender 400 (Seward Medical, London, UK). Serial dilutions of the homogenates were placed on plate count agar (PCA; Biokar Diagnostics. Beauvais, France) and incubated at 35 ± 1 °C for 48 h for TBC determination. The results were expressed as log CFU g^1^ [36].

### 2.6. Respiratory Activity

A Micro-GC gas analyser (Model CP 2002, Chrompack International, Middelburg, The Netherlands) characterised by a thermal conductivity detector was applied to evaluate the respiratory activity of carrot pieces. A variation of the procedure reported by López-Gómez et al. [37] was used. At each monitoring time, 50 g of carrots were placed in airtight containers of 250 mL and stored at 4 °C for 3 h. Subsequently, 1.7 mL of gas was collected from the headspace with a syringe via a rubber septum. Carbon dioxide production (RRCO_2_) and oxygen consumption (RRO_2_) were reported as mg kg^−1^ h^−1^, as described by Tappi et al. [38].

### 2.7. Extraction and Determination of Total Phenolic and Content and Antioxidant Activity

The extraction of phenolic compounds was carried out as reported by Formica-Oliveira et al. [39], with minor modifications. A portion of 5 g of carrot was homogenised with 20 mL of methanol with an Ultra-Turrax T25 (IKA^®^-Werke GmbH & Co., Staufen, Germany) for 2 min. Homogenates were centrifuged at 13,500× *g* and 4 °C for 20 min (Centrifuge AVANTITM J-25, Beckman Instruments Inc., Fullerton, CA, USA). The supernatants were collected and then filtered through Whatman no. 1 filter.

The total phenolic content (TPC) was determined following the Folin–Ciocalteu procedure tailored to 96-well microplates [2]. After an aliquot of 30 µL of methanolic extract was introduced into a microplate, 150 μL of 10% (*v*/*v*) Folin–Ciocalteu reagent and 120 μL of Na_2_CO_3_ 7.5% (*w*/*v*) were added. After incubating for 90 min at room temperature in darkness, the absorbance at 765 nm was determined with a microplate reader (Thermo Scientific Multiskan GO, Vantaa, Finland). The results were expressed as mg of gallic acid equivalents per 100 g (mg 100 g^−1^) on a fresh-weight basis. Phenolic compounds extraction was carried out twice per treatment repetition and fourfold spectrophotometrically determined.

Total carotenoids content (TCC) was determined spectrophotometrically (CECIL CE 2021; Cecil Instruments Ltd., Cambridge, UK) as described by González-Casado et al. [40], with slight modifications. An aliquot of fresh-cut carrots (2 g) was homogenised with 25 mL of acetone:ethanol (1:1, *v*/*v*) with an Ultra-Turrax T25 (IKA^®^ WERKE, Germany). Sample was extracted in the dark, filtered through Whatman No. 4 filter paper, and washed with the acetone:ethanol solution until the residue was colourless. Samples were adjusted to 100 mL, and the absorbance was read at 470 nm versus a blank of acetone:ethanol. TCC was determined by the equation below:(3)$$\mathrm{Total}\;\mathrm{Carotenoids}\;\mathrm{Content}=\frac{A_{470}\cdot V\cdot 10^{4}}{A_{1cm}^{1\%}\cdot G}$$
where *A*_470_ is the absorbance at 470 nm, *V* is the total volume of extract (mL), $A_{1cm}^{1\%}$ is the extinction coefficient of a mixture of carotenoids established as 2500 by Gross [41] and *G* is the sample weight (g). Total carotenoids were expressed as mg per 100 g of fresh weight (mg TCC 100 g^−1^).

The antioxidant activity of fresh-cut carrots was determined by DPPH and ABTS assays, using a 96-well microplate reader (Thermo Scientific Multiskan GO, Vantaa, Finland).

The determination of free radical scavenging effect on 1,1-diphenyl-2-picrylhydrazyl (DPPH) radical was performed as reported by Ribas-Agustí et al. [35] with some modifications. An aliquot of 20 µL of methanolic extract (Section 2.7) was placed into a microplate and 280 μL of a 6 × 10^−5^ M methanolic solution of DPPH were added. The homogenate was kept in darkness for 30 min under continuous stirring and the absorption of the samples was measured at 515 nm against a blank of methanol without DPPH. A calibration curve was built with Trolox (from 6 to 30 μM). Results were expressed as µM Trolox equivalents per 100 g (µmol 100 g^−1^) on a fresh-weight basis.

The ABTS (2,2’-azino-bis acid 3-ethylbenzothiazolin-6-sulfonic acid) assay was carried out as reported by Re at al. [42]. An aliquot of 40 µL of methanolic extract (Section 2.7) was placed into a microplate and 260 μL of ABTS ethanol solution were added. The homogenate was kept in darkness for 6 min under continuous stirring and the absorption of the samples was measured at 734 nm against a blank of ethanol without ABTS. A calibration curve was built with Trolox (from 30 to 120 μM). The results were expressed as µM Trolox equivalents per 100 g (µmol 100 g^−1^) on a fresh-weight basis.

### 2.8. Statistical Analysis

All the experimental results were expressed as mean value (*n* = 4) ± standard deviation (mean ± SD). Significance of the results and statistical differences were analysed using SPSS Software (Version 15.0, SPSS Inc., Chicago, IL, USA). One-way analysis of variance (ANOVA) and several multiple comparisons, by Tukey’s post-hoc test, were conducted to identify individual significant differences (*p* < 0.05). The Pearson’s correlation test was performed to determine correlation coefficients (r) among polyphenolic compounds and antioxidant assays.

## 3. Results and Discussion

### 3.1. Physicochemical Properties

Physicochemical characterisation results of uncoated (W and W + LE) and coated (PC and PC + LE) fresh-cut carrots are reported in Table 1.

The application of a pectin-based coating containing lemon byproduct extract (PC + LE) on minimally processed carrots helped to preserve their physiological parameters, helping to the maintain higher total soluble solids and acidity values during refrigerated storage at 4 °C. This is suggestive of reduced physiological activity in the coated product [43]. Immediately after treatment, PC + LE carrots showed the lowest acidity values, compared to other fresh-cut carrots. The combination of the lemon byproduct extract and pectin-based coating favoured a relevant preservation of the acidity levels, showing no significant changes (*p* > 0.05) throughout storage. In addition, PC + LE carrots exhibited the highest soluble solids content (°Brix) from the first to the last day of storage, with a very slight decreasing trend over time. Physiological parameters, such as titratable acidity, pH and sugar content, are good indicators of fruit maturation and senescence. In this regard, the coatings successfully demonstrated their potential in slowing down the natural physiological behaviour of fruits and vegetables, retarding metabolic reactions and, thus, senescence, thanks to the different gas permeability of the coating and its influence on the vegetables’ respiratory activity [6,44]. Moreover, thanks to its content in bioactive compounds, the simultaneous presence of LE also contributed to delaying changes induced by stressful conditions concomitant with minimal carrots processing, such as the metabolism of soluble sugars.

Colour changes were monitored over time by determining CIELab parameters (lightness, L *; green-red chromaticity, a *; and blue-yellow chromaticity, b *) and total colour difference (Table 2), on both cortical tissue and vascular cylinder of uncoated and coated fresh-cut carrots. Total colour difference (ΔE) values throughout storage were significantly (*p* < 0.05) reduced by the application of a pectin-based coating and a lemon byproduct extract.

As shown in Table 2, coated carrots were characterized by the lowest ΔE values on both cortical tissue and vascular cylinder immediately after treatment (day 1). Moreover, the combination of the pectin coating and lemon byproducts extract (PC + LE) limited colour changes throughout storage (ΔE < 3). It is remarkable that total-colour-difference values greater than 3 denote differences that are easily noticeable by the human eye [45]. In contrast, uncoated carrots underwent colour changes (ΔE > 3) over storage that are noticeable by the human eye in both cortical and vascular cylinder tissues. Interestingly, W + LE-treated carrots were characterised by significantly lower ΔE values (*p* < 0.05) compared to W-treated carrots, which can be attributed to the antibrowning effect of the lemon extract.

The coating application was related to a significant (*p* < 0.05) greater stability of lightness (L *) values and with higher a * and b * values throughout storage in comparison to uncoated carrots (data not reported). The higher L * values observed in W and W + LE carrots could be related to the biosynthesis of lignin in wounded carrot tissues by enzymes stimulated during minimal processing operations, such as phenylalanine ammonia lyase (PAL) [46]. As reported by several authors [47,48,49,50], PAL promotes surface discoloration by increasing levels of soluble phenolics required for lignin biosynthesis and its activity could increase due to wound-induced stimulation throughout storage.

In this study, the presence of the pectin-based coating and LE on carrots’ surfaces appears to have contributed to the retention of colour over time on both the cortical tissue and vascular cylinder, by counteracting the enzymatic action stimulated by abiotic stress.

As shown in Figure 1, coating treatments significantly (*p* < 0.05) affected the development of white blush on fresh-cut carrots during storage. The WI values of coated carrots were significantly lower than those of the uncoated ones, throughout the evaluated storage period. In all the fresh-cut carrots, vascular cylinder surfaces exhibited higher WI values than cortical surfaces over time, which is consistent with the observed ΔE values (Table 2). PC and PC + LE carrots did not suffer an increase in WI throughout storage, recording constant values on both cortical tissue and vascular cylinder. Comparable findings were also noted by Vargas et al. [51] for carrot slices, who reported that a chitosan coating significantly reduced the development of white blush on carrots’ surfaces. In contrast, an increase in WI values was observed in the vascular cylinder surfaces of uncoated carrots (W and W + LE) from the third day of storage onwards. Moreover, the addition of the lemon byproduct extract into the pectin-coating formulation did not show any influence on the development of white blush throughout storage.

In this study, the increase in white-blush development was significantly (*p* < 0.05) retarded by the application of the pectin-based coating. Considering that the main reason for carrot white discoloration is surface dehydration, the edible coating delayed whitening by working as a surface moisturizer [52,53]. Hence, acting as a humectant thanks to its hydrophilic nature, pectin keeps the surface of peeled carrots moist to retard white discolouration (Figure 2).

Regarding texture (Figure 3), the coated fresh-cut carrots showed similar (*p* > 0.05) hardness values in both cortical tissue and vascular cylinder throughout storage. This contrasts with uncoated carrots, whose texture values significantly (*p* < 0.05) differed, with vascular cylinders softer than cortical tissues. This difference between tissues, observed from day 1 in W and W + LE carrots, was consistent throughout storage.

The use of calcium chloride in the coating formulation in combination with LE seems to be a determinant factor in maintaining the hardness of fresh-cut carrots. Various authors described the positive impact of incorporating calcium chloride into coating formulations on the retention of hardness in fresh-cut fruits [15,54,55,56]. However, the presence of LE helped, as well, to preserve the hardness of carrots, probably thanks to the extract’s action in counteracting the activity of specific enzymes such as polygalacturonase. Hardness loss, in fact, may be related with the deterioration of compounds responsible for vegetable structural rigidity, primarily insoluble pectin and protopectin. In maturation, pectinesterase and polygalacturonase activities intensify, producing the solubilisation of pectin substances [57]. In addition, the observed decrease in hardness of the fresh-cut carrots may be linked not only to the action of pectinolytic enzymes, but also to the increased activity of glycolytic enzymes, which contribute to the hydrolysis of hemicellulose and other cell wall components and which might be activated as a defence mechanism in cases of microbiological attack and/or injury [58]. The structural integrity of carrot tissues could also be related to the highest soluble solids content observed for PC + LE carrots (Table 2). In fact, although soluble sugars are known to be a signal of the regulation of various mechanisms associated to growth, development and metabolic responses in plants, they also operate as metabolic resources and structural elements of cells [59].

### 3.2. Microbial Growth

The results of microbiological analyses are reported in Figure 4. Microbial counts on PC + LE significantly differed from those observed in carrots subjected to other treatments. PC + LE carrots showed the lowest value of TBC (2.58 ± 0.06 log CFU g^−^^1^) immediately after treatment. Microbial counts for treatments without the incorporation of LE were significantly (*p* < 0.01) higher (3.66 ± 0.31 and 3.81 ± 0.44 log CFU g^−^^1^, for W and PC carrots, respectively). After the first day of storage, PC + LE continued to exhibit the lowest TBC values (*p* < 0.01) for at least one week. Our results agree with those obtained by Amanatidou et al. [60], who studied the effect of an alginate-based coating added with 0.1% of citric acid on sliced carrots stored at 8 °C under modified atmosphere conditions. The authors reported that the combination of 0.1% citric acid and 2% CaCl_2_ in coating formulations significantly reduced the initial total microbiota for at least 1 log CFU g^−^^1^ during up to 8 days of storage.

The inclusion of natural antioxidants and antimicrobials to the pectin-based coating, by means of the incorporation of LE, decisively led to the shelf-life extension of fresh-cut carrots. This effect could be related to the presence of phenolic compounds dispersed within the pectin matrix of the coating, which allowed them to be progressively released to the carrot surface over storage [61]. As reported by Budiati et al. [62] and Ivasenko et al. [63], compounds such as apigenin and gallic acid, detected in lemon byproduct extracts, are characterised by an intense antibacterial activity, by limiting microbial adhesion and deactivationg bacterial enzymes and cell transport proteins. In addition, hardness decay may be, as well, related to the proliferation of pectolytic Pseudomonas [60]. In fact, the higher and more stable texture values observed for PC + LE (Figure 3) correlate well with lower total bacterial counts, which could be associated to reduced bacterial enzymatic activities.

### 3.3. Respiratory Activity

The effect of treatments on the respiratory activity of fresh-cut carrots, expressed as oxygen consumption (RRO_2_) and carbon dioxide production (RRCO_2_), is shown in Figure 5.

Significant differences (*p* < 0.05) in terms of O_2_ consumption and CO_2_ production were found between treated carrots immediately after treatment. Particularly, coated fresh-cut carrots (PC and PC + LE) showed the lowest values of RRO_2_ (Figure 5a). Furthermore, an increasing trend was observed for RRO_2_ values over storage, with the highest values observed for W-treated carrots. The reduced oxygen consumption in coated fresh-cut carrots is probably attributable to the slowing down of the product’s metabolic reactions, which would result in a better preservation of physicochemical parameters. On the other hand, regarding RRCO_2_ (Figure 5b), all treatments showed an initial increase in values followed by a decrease after the 7th day of storage.

Variations in respiration rates furnish information of the general metabolic activity of carrot tissues as affected by postharvest and minimal processing conditions [37]. As reported by Tappi et al. [38], the inhibition of respiration relative to O_2_ consumed did not always match a significant decrease in CO_2_ production (RRCO_2_). This contrasting behaviour could be caused by the stress generated during processing and coating operations, which promoted a slight increase in CO_2_ production. However, the application of the lemon byproduct extract in the coating formulation does not appear to have had any significant impact on the respiration rate of fresh-cut carrots.

In addition, the coating itself could behave as a barrier to gas transport and prolong the commercial shelf-life of fresh products by modifying their internal atmosphere [51]. At the same time, the increase in the respiration process observed in coated carrots could be associated to modifications in the barrier properties of the coating matrix, which may be stimulated by the high values of carrots’ water activity when stored in chilling conditions [64]. The changes in respiratory rates could also be linked to vegetable-tissue stress induced by minimal processing operations, such as trimming, peeling and cutting, or the application of the coating matrix, which generates stressful conditions [4]. However, the presence of the pectin-based coating encouraged a significant reduction in the respiration rate of fresh-cut carrots for at least one week of storage compared to the control W, most likely as a consequence of the interaction of calcium ions with pectin and plant-cell-wall components.

### 3.4. Bioactive Compounds Content and Antioxidant Activity

The total carotenoids content (TCC) of fresh-cut carrots, as affected by the coating treatments, is reported in Figure 6. The highest total carotenoid content was found in just-processed carrots subjected to treatments with the incorporation of LE, with values of 16.83 ± 0.67 and 13.29 ± 0.75 mg TCC 100 g^−^^1^ for W + LE and PC + LE, respectively. Among them, PC + LE showed constant values in TCC for one week of storage. This aspect confirms the simultaneous protective effect of the coating and the extract on the stability of carotenoids over time, a parameter that could be related to the minimal colour variation found in PBC + LE (Table 2). Changes in colour can be linked to dehydration and discolouration of the carrots’ surfaces, together with the oxidation of carotenoids [51]. In fact, the main cause of carotenoid losses in vegetables is the oxidation of carotenoids, due to their highly unsaturated structure, which can occur by the spontaneous reaction of vegetable tissues in the presence of oxygen and other environmental factors such as light [65]. After the first week, a progressive decrease in TCC was observed until the last day of storage, except for W and PC, which showed an increment in total carotenoid content after the 10th day of storage.

Total phenolic content (TPC) and antioxidant activity values (Figure 7) exhibited a similar trend to that observed for TCC, especially in fresh-cut carrots with added LE. The incorporation of the lemon byproduct extract into the pectin-based coating formulation assured a greater stability of the bioactive composition of the carrots during storage. TPC values clearly denote a protecting effect of the coating against oxidative phenomena [61]. Hence, treatments incorporating the lemon byproduct extract (PC + LE and W + LE) allowed maintaining a higher total polyphenols content than control samples (W and PC), until the 10th day of storage. This contrasts with the values observed for W and PC carrots, which noticeably decreased during the first week of storage. However, on the 10th day, treatments without the incorporation of LE showed a significant increase in TPC as well as antioxidant activity values, as also observed by Ranjitha et al. [66].

This phenomenon may be ascribed to the onset of metabolic pathways leading to the production of phenolic compounds, as a part of the plant defence mechanisms [61]. An increase in TPC is one of the most widely studied events in response to wounding in several fresh-cut products, which has been further confirmed in the case of carrots. Namely, an increase in TPC and TCC might also be related to the wound-induced stimulation of the plant enzyme phenylalanine ammonia lyase (PAL), which transforms L-phenilalanine into trans-cinnamic acid, which acts as a precursor for different phenylpropanoids, such as lignin [66]. In this regard, several authors assign development of a superficial white blush to lignin synthesis in reply to cellular injuries, where lignin performs as a new barrier [51,67,68]. This is consistent with total-colour-difference (Table 2) and whiteness-index (Figure 1) results. However, the application of the edible coatings added with LE was effective in preserving the the TPC and antioxidant activity of coated carrots, which is related to the contents of different compounds, such as polyphenols and carotenoids. A high positive correlation between TCC and DPPH assays was found for W + LE carrots (r = 0.90) and PC + LE (r = 0.83), which agrees with the well-established relationship between the protective affect exerted by the lemon byproduct extract on carotenoids. As reported by de Oliveira et al. [43], the application of coatings could reduce the loss of phenolic compounds and preserve coated fruit and vegetables from oxidative damage and the accumulation of free radicals, thanks to their contribution in slowing down enzymatic activity and, thus, retarding changes in colour and other physicochemical parameters associated to senescence [69].

## 4. Conclusions

The results of this study demonstrate that a pectin-based edible coating could be a simple and affordable technique for carrying bioactive compounds, such as those contained in the lemon byproduct extract, with considerable applicability in minimally processed carrots to improve the nutritional value and quality attributes of these fresh goods.

In fact, fresh-cut carrots treated with a pectin-based coating and lemon byproduct extract were characterized by a good preservation of physiological parameters and limited changes in colour (ΔE < 3) and white-blush development on both cortical tissue and vascular cylinder throughout the storage period. The application of the pectin-based coating with the lemon byproduct extract ensured that carrots maintained stable structural integrity throughout the 14 days of storage at 4 °C, thanks to the reduction in enzymatic bacterial activity. This kind of treatment on minimally processed carrots also resulted in higher levels of carotenoids, phenolic compounds and antioxidant activity, as evident from higher ABTS and DPPH radical scavenging activity values.

This strategy represents an interesting option to decrease the rate of the physiological postharvest decay of fresh-cut carrots, slowing down the respiration process and senescence throughout storage and, at the same time, could be considered a valid approach to valorise a food industry byproduct. In this regard, edible coatings have proven to be promising carrier systems for bioactive ingredients that can improve food functional properties, representing the ideal choice for fruit and vegetables to maintain quality attributes for a longer shelf-life.

## Figures and Tables

**Figure 1 foods-11-01314-f001:**
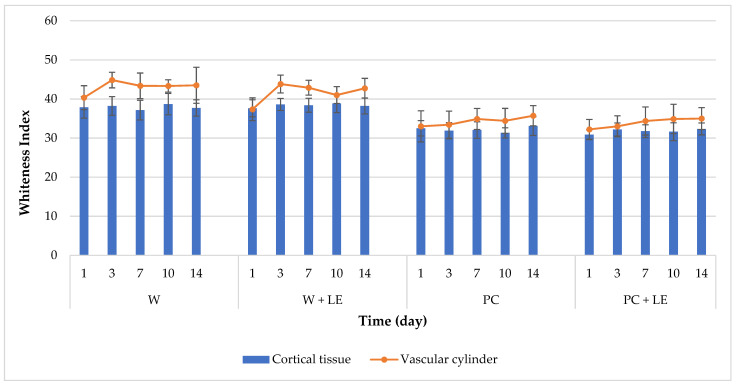
Changes in whiteness index (WI) values of uncoated and coated fresh-cut carrots on cortical tissue and vascular cylinder throughout 14 days of storage (4 °C). Abbreviations: W; W + LE; PC; PC + LE (see Table 1).

**Figure 2 foods-11-01314-f002:**
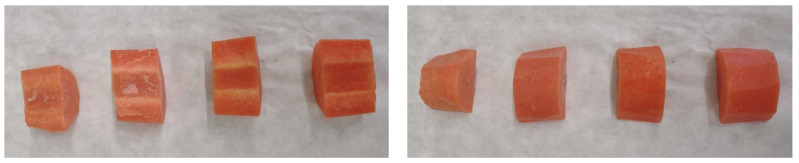
Representative photograph of white blush in vascular cylinder (**on the left**) and cortical tissue (**on the right**) of uncoated and coated fresh-cut carrots on the 14th day of storage at 4 °C. From left to right: W; W + LE; PC; PC + LE (abbreviations, see Table 1).

**Figure 3 foods-11-01314-f003:**
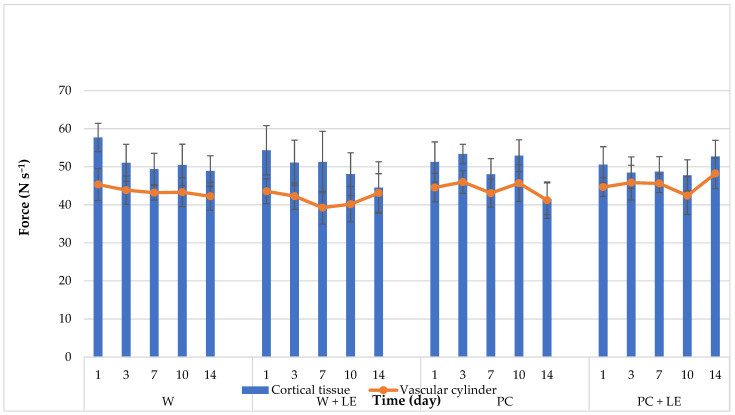
Hardness changes throughout storage of fresh-cut carrots (cortical and vascular cylinder tissues) as affected by pectin coating and the addition of a lemon byproduct extract. Abbreviations: W; W + LE; PC; PC + LE (see Table 1).

**Figure 4 foods-11-01314-f004:**
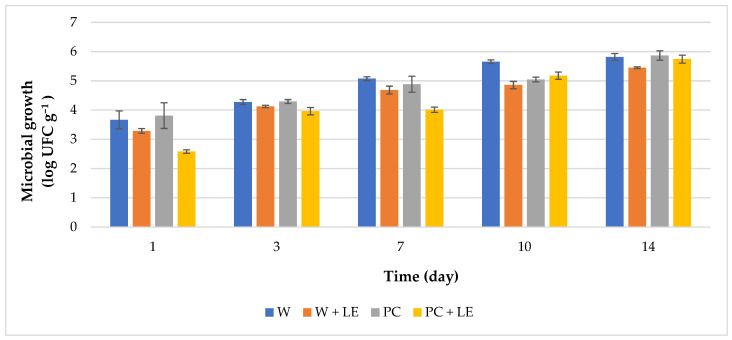
Growth of total bacterial load on uncoated and coated fresh-cut carrots throughout 14 days of storage (4 °C). Abbreviations: W; W + LE; PC; PC + LE (see Table 1).

**Figure 5 foods-11-01314-f005:**
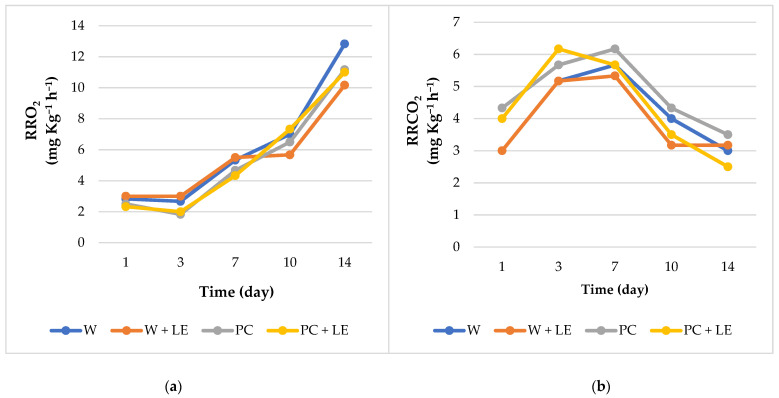
(**a**) Oxygen consumption values (RRO_2_) of uncoated and coated fresh-cut carrots throughout 14 days of storage at 4 °C; (**b**) carbon dioxide production values (RRCO_2_) of uncoated and coated fresh-cut carrots throughout 14 days of storage at 4 °C. Abbreviations: W; W + LE; PC; PC + LE (see Table 1).

**Figure 6 foods-11-01314-f006:**
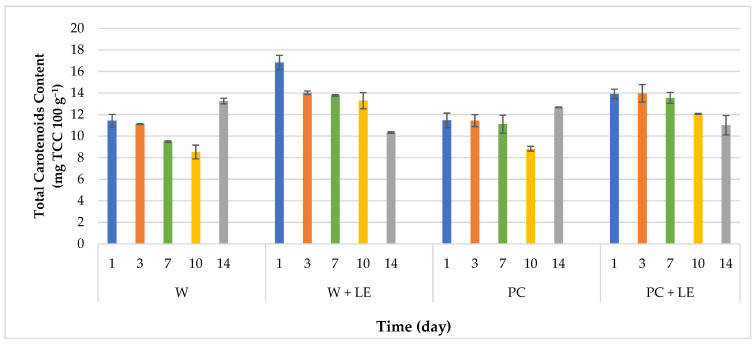
Total carotenoids content of uncoated and coated fresh-cut carrots throughout 14 days of storage (4 °C). Abbreviations: W; W + LE; PC; PC + LE (see Table 1).

**Figure 7 foods-11-01314-f007:**
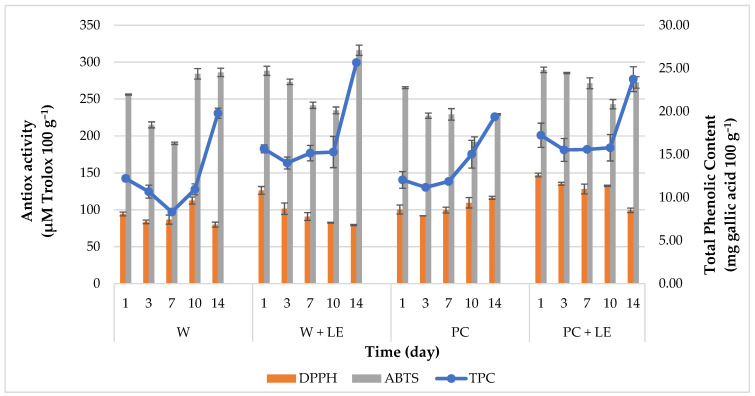
Total phenolic content (TPC) and expression of antioxidant activity (ABTS and DPPH assays) of uncoated and coated fresh-cut carrots throughout 14 days of storage (4 °C). Abbreviations: W; W + LE; PC; PC + LE (see Table 1).

**Table 1 foods-11-01314-t001:** Changes in acidity, pH, total soluble solids (Brix°) and dry matter of uncoated and coated fresh-cut carrots during 14 days of storage (4 °C).

	Storage Time (Day)				
	1	3	7	10	14
Acidity	Mean ± SD	Mean ± SD	Mean ± SD	Mean ± SD	Mean ± SD
W	0.10 ± 0.00 ^ab,A^	0.07 ± 0.01 ^C^	0.07 ± 0.01 ^b,BC^	0.08 ± 0.01 ^b,B^	0.10 ± 0.00 ^a,A^
W + LE	0.10 ± 0.01 ^ab,A^	0.07 ± 0.00 ^BC^	0.11 ± 0.00 ^a,A^	0.08 ± 0.00 ^b,AB^	0.05 ± 0.02 ^b,C^
PC	0.11 ± 0.00 ^a,A^	0.08 ± 0.01 ^B^	0.08 ± 0.00 ^b,B^	0.12 ± 0.03 ^a,A^	0.07 ±0.00 ^ab,B^
PC + LE	0.09 ± 0.01 ^b^	0.07 ± 0.01	0.09 ± 0.01 ^a,b^	0.07 ± 0.01 ^b^	0.09 ± 0.02 ^a^
Significance	*	ns	**	**	**
pH					
W	6.34 ± 0.04 ^BC^	6.32 ± 0.03 ^C^	6.43 ± 0.01 ^AB^	6.6 ± 0.03 ^a,A^	6.49 ± 0.10 ^a,B^
W + LE	6.27 ± 0.09	6.46 ± 0.10	6.54 ± 0.08	6.51 ± 0.25 ^a^	6.43 ± 0.00 ^a^
PC	6.31 ± 0.05 ^A^	6.38 ± 0.01 ^A^	6.40 ± 0.06 ^A^	6.04 ± 0.26 ^b,B^	6.21 ± 0.00 ^b,AB^
PC + LE	6.31 ± 0.01 ^C^	6.29 ± 0.16 ^C^	6.52 ± 0.13 ^B^	6.73 ± 0.06 ^a,A^	6.24 ± 0.06 ^b,C^
Significance	ns	ns	ns	**	**
Soluble solids content (°Brix)					
W	3.6 ± 0.14 ^d^	3.25 ± 0.07 ^c^	3.60 ± 0.57 ^c^	3.70 ± 0.28 ^c^	3.90 ± 0.99 ^b^
W + LE	5.65 ± 0.21 ^b,A^	5.20 ± 0.42 ^b,AB^	5.00 ± 0.42 ^b,B^	5.65 ± 0.07 ^b,A^	5.60 ± 0.28 ^a,A^
PC	5.35 ± 0.07 ^c^	5.40 ± 0.14 ^b^	4.90 ± 0.42 ^b^	5.30 ± 0.28 ^b^	5.35 ± 1.34 ^a,b^
PC + LE	6.4 ± 0.14 ^a,A^	6.05 ± 0.07 ^a,C^	6.25 ± 0.07 ^a,ABC^	6.35 ± 0.21 ^a,AB^	6.10 ± 0.14 ^a,BC^
Significance	**	**	**	**	**
Dry matter					
W	9.24 ± 0.18 ^b^	9.94 ± 0.94 ^a^	9.99 ± 1.15	9.60 ± 0.45 ^ab^	9.50 ± 0.07
W + LE	10.47 ± 0.78 ^a^	9.71 ± 0.21 ^a^	10.48 ± 1.76	10.11 ± 0.32 ^a^	9.19 ± 0.41
PC	10.96 ± 0.50 ^a,A^	9.73 ± 0.10 ^a,B^	9.18 ± 0.18 ^B^	9.41 ± 0.03 ^b,B^	9.31 ± 0.84 ^B^
PC + LE	9.20 ± 0.36 ^b^	8.68 ± 0.28 ^b^	8.73 ± 0.68	9.55 ± 0.50 ^ab^	9.22 ± 1.53
Significance	**	**	ns	*	ns

Values are the mean of four determinations ± SD (*n* = 4). Different superscript letters within a row or column indicate statistically significant differences between uncoated and coated fresh-cut carrots. Lower case letters denote differences among treatments for a set treatment time. Upper case letters denote differences among treatment times for a set treatment. Absence of letters indicate non statistically significant differences. ** Significance at *p* < 0.01; * Significance at *p* < 0.05; ns, not significant. W, carrots dipped in water; W + LE, carrots dipped in water + LE solution; PC, carrots coated with a pectin-based coating; PC + LE, carrots coated with a pectin-based coating with LE.

**Table 2 foods-11-01314-t002:** Total colour difference (ΔE) of uncoated and coated fresh-cut carrots throughout 14 days of storage (4 °C).

ΔE										
	Storage Time (Day)									
	1		3		7		10		14	
	Cortical Tissue	Vascular Cylinder	Cortical Tissue	Vascular Cylinder	Cortical Tissue	Vascular Cylinder	Cortical Tissue	Vascular Cylinder	Cortical Tissue	Vascular Cylinder
W	3.63 ± 0.56 ^a^	7.09 ± 0.53 ^ab,B^	5.85 ± 1.24 ^a^	11.31 ± 1.48 ^a,A^	5.60 ± 1.27 ^a^	9.67 ± 1.42 ^a,A^	5.83 ± 2.04 ^a^	9.84 ± 1.52 ^A^	5.47 ± 0.94 ^a^	9.62 ± 1.72 ^a,A^
W + LE	3.01 ± 0.30 ^ab^	6.49 ± 1.35 ^a,B^	2.33 ± 0.93 ^b^	13.87 ± 2.00 ^a,A^	2.87 ± 0.04 ^b^	9.01 ± 1.88 ^a,AB^	3.53 ± 1.26 ^a,b^	7.89 ± 2.35 ^B^	2.33 ± 0.91 ^b^	9.57 ± 1.67 ^a,AB^
PC	2.20 ± 0.64 ^bc^	3.17 ± 0.56 ^ab^	2.07 ± 0.56 ^b^	6.21 ± 0.70 ^b^	1.42 ± 0.61 ^b^	4.41 ± 1.26 ^b^	1.66 ± 0.58 ^b^	5.93 ± 2.58	1.63 ± 0.69 ^b^	6.14 ± 2.06 ^a,b^
PC + LE	1.63 ± 0.19 ^c,B^	1.74 ± 0.66 ^b,B^	3.06 ± 0.26 ^b,A^	2.07 ± 0.71 ^c,AB^	1.00 ± 0.28 ^b,B^	3.02 ± 0.19 ^b,AB^	1.22 ± 0.07 ^b,B^	4.38 ± 1.78 ^A^	2.65 ± 0.36 ^b,A^	3.76 ± 0.28 ^b,AB^
Sign.	**	*	**	**	**	**	**	ns	**	**

Values are reported as mean ± standard deviation (*n* = 10). Different superscript letters within a row or column indicate statistically significant differences between uncoated and coated fresh-cut carrots. Lower case letters denote differences among treatments for a set treatment time. Upper case letters denote differences among treatment times for a set treatment. Absence of letters indicate non statistically significant differences. Abbreviations: **; *; ns; Sign.; W; W + LE; PC; PC + LE (see Table 1).

## Data Availability

Data is contained within the article.

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
