# Peer review of "Efficacy of Pectin-Based Coating Added with a Lemon Byproduct Extract on Quality Preservation of Fresh-Cut Carrots"

_foods, 2022, doi:10.3390/foods11091314_

Round 1

Reviewer 1 Report

Dear Editor/Author,

A manuscript: Imeneo, V., Romeo, R., De Bruno, A., Piscopo, A. (2022).  Green-sustainable extraction techniques for the recovery of antioxidant compounds from “citrus Limon” by-products. Journal of Environmental Science and Health, Part B. https://doi.org/10.1080/03601234.2022.2046993

shows the techniques for obtaining extracts from citrus lemon by-products and their characterization. In this paper one of the obtained extracts was used for activation of pectin biopolymer films and applied on fresh carrots as a continuation of author’s research.

The title and the abstract correspond to the set task and goal of the paper. Literary quotations are contemporary, from the field from which the topic of this paper is. English is correct, the applied terminology good, stylistic and linguistic errors were not noticed. Selected methods of monitoring the quality of the tested carrots are sufficient to present the effect of the applied coating with and without lemon extract. The graphical, tabular and textual presentation of the results is clear and unambiguous. What, in my opinion, gives the greatest value to the work is the excellent discussion of the results in the sense that for each examined characteristic an explanation is given with the underlying mechanism.

Author Response

Thank you so much for your revision and comments. We are submitting an improved version of the manuscript considering all the issues raised by the reviewers.

Reviewer 2 Report

The title needs to be modified.

Why are the quality parameters of carrots examined in just 14 days of storage? Be sure to mention the shelf life of uncoated carrots naturally at refrigerator temperature with valid reference.

The authors should mention the reason for selecting this hydrocolloid, carrot and lemon extracts clearly and explains the objective of this study and its impact.

The abstract should be more informative by giving real results rather than elastic sentences. Important and main contents should be given. Support the results with some quantitative data. Moreover, no conclusions are provided.

The introduction is very incomplete and desperately needs to be rewritten and seriously revised.

Line 89: What is the reason for using low-methoxyl pectin?

Line 119: Why did you choose only 1% lemon extract? Do not different concentrations affect the shelf life of the carrot? Give a compelling reason. How is a constant concentration of 2% pectin or a constant concentration of 1.5% glycerol obtained? How can you prove that it was the most appropriate concentration?

Line 140: To measure carrot dry matter, refer to the standard, not Piscopo et al. Change the reference of this section.

Line 195: There is no title to determine Total carotenoids content.

Summarize the contents of titles 2.7 and 2.8.

Mention meaningful letters in all figures.

Be sure to use the article “Modeling the respiration rate of chitosan coated fresh in-hull pistachios (Pistacia vera L. cv. Badami) for modified atmosphere packaging design” to complete analyzes related to Respiratory activity.

Author Response

Dear Reviewer,

Thank you so much for you revision and suggestions.

Below are the replies to your comments.

  • The title needs to be modified.

We thank you for your suggestions. We have modified the title to better fit it with the research topic.

  • Why are the quality parameters of carrots examined in just 14 days of storage? Be sure to mention the shelf life of uncoated carrots naturally at refrigerator temperature with valid reference.

Concerning the choice to determine the quality parameters of the carrot samples for 14 days, a previous determination of the respiration rate of a commercial sample of ready-to-eat carrots close to the end of its shelf-life (about 10 days after packaging) showed higher oxygen consumption and carbon dioxide production values than those found in our study at the same monitoring time. This result was interpreted as an excessive vegetable stress condition and for this reason it was considered appropriate to focus the characterisation of the samples over a period of two weeks. In addition this is a very common temporary frame when considering the shelf-life of minimally processed products.

  • The authors should mention the reason for selecting this hydrocolloid, carrot and lemon extracts clearly and explains the objective of this study and its impact.

The choice of pectin as the hydrocolloid for the formulation of the edible coating was dictated by the results obtained from a previous screening in which several food grade polysaccharides, such as pectin, alginate, gellan gum and chitosan, were tested on fresh cut carrots. At the end of the screening, pectin was chosen as the most appropriate hydrocolloid.

Moreover, carrots are a widely consumed product, available throughout the year, suitable for all categories of consumers and rich in useful compounds (as stated in the introduction to the manuscript). At the same time, one of the main problems with ready-to-eat carrots is the development of white blush on the surface, which reduces their acceptability to consumers and contribute to the reduction of the shelf-life. With the application of the enriched coating, the aim was to reduce this quality defect of the carrots as much as possible and, at the same time, to guarantee a greater contribution in useful compounds with antioxidant activity provided by the addition of the lemon peel extract.

Finally, the choice of lemon by-product extract is clearly explained in a previous study (Imeneo, V., Romeo, R., De Bruno, A., Piscopo, A., 2022.  Green-sustainable extraction techniques for the recovery of antioxidant compounds from “citrus Limon” by-products. Journal of Environmental Science and Health, Part B. https://doi.org/10.1080/03601234.2022.2046993), of which this is a subsequent application.

  • The abstract should be more informative by giving real results rather than elastic sentences. Important and main contents should be given. Support the results with some quantitative data. Moreover, no conclusions are provided.
  • The introduction is very incomplete and desperately needs to be rewritten and seriously revised.

We thank for your suggestions. We have improved the abstract and introduction sections.

  • Line 89: What is the reason for using low-methoxyl pectin?

Line 89: The reason for using low-methoxyl pectin is explained in the introduction section (lines 59-61) and supported by bibliographical references [14; 15; 16]. The choice is dictated by the fact that this type of polysaccharide is generally used in coating formulations due to its ability to form strong gels structures or insoluble polymers in the presence of multivalent metal cations. In this study, in fact, a calcium chloride solution was also used in the coating formulation for pectin crosslinking.

  • Line 119: Why did you choose only 1% lemon extract? Do not different concentrations affect the shelf life of the carrot? Give a compelling reason. How is a constant concentration of 2% pectin or a constant concentration of 1.5% glycerol obtained? How can you prove that it was the most appropriate concentration?

Line 119: the concentration of the lemon by-product extract (1%) was chosen based on what is reported in the literature, since it has been seen in many studies that an extract concentration higher than 3% is not used to formulate an enriched coating. For this reason, a preliminary test was conducted on the formulation of the pectin coating and its application on fresh cut carrots, considering different concentrations of lemon by-product extract (1% - 1.5% - 3%). It is true that different concentrations can influence the shelf life of carrots, but it is not always the case that a higher concentration is the best for extending shelf life. In fact, as reported in our study, an addition of external substances on minimally processed fruit and vegetables could cause an increase in stress conditions and affect the shelf-life of the product.

Moreover, coating formulation was prepared following what reported by Oms-Oliu, G.; Soliva-Fortuny, R.; Martín-Belloso, O. Edible coatings with antibrowning agents to maintain sensory quality and antioxidant properties of fresh-cut pears. Postharvest Biol. Technol. 2008, 50(1), 87–94. 541

https://doi.org/10.1016/j.postharvbio.2008.03.005

In this study the authors proved that 2% pectin and 1.5% glycerol concentrations were the optimal one for edible coating formulations with different typologies of polysaccharides.

  • Line 140: To measure carrot dry matter, refer to the standard, not Piscopo et al. Change the reference of this section.

It was modified as suggested.

  • Line 195: There is no title to determine Total carotenoids content.

It was modified as suggested.

  • Summarize the contents of titles 2.7 and 2.8.

Titles sections 2.7 and 2.8: it was modified as suggested.

  • Mention meaningful letters in all figures.

 We did not include letter stating significant differences in the figures as comparison between treatments and storage times made it difficult to get a clear sight of the statistical differences. Instead, it is mentioned within the text when significant differences were noticed.

  • Be sure to use the article “Modelling the respiration rate of chitosan coated fresh in-hull pistachios (Pistacia vera L. cv. Badami) for modified atmosphere packaging design” to complete analyses related to Respiratory activity

The manuscript was considered and cited as requested. Thank you.

Reviewer 3 Report

The manuscript received for review develops new type of pectin-based coating with added lemon extract for fresh cut carrot preservation. This topic is relevant for the journal scope, while authors try to utilize lemon by-products (or waste products) and its’ phenolic and antioxidant compounds for positive effect on carrots’ shelf-life extension.

Manuscript contains enough significant original material and it is clearly and concisely written.

The title of the manuscript is informative, corresponds to the content of the work, but need some rephrasing, to be clearer.

Introduction section comprehensive and sufficient.

The Materials and Methods section describes all conducted testing. Regarding microbiological testing, there is question why only one parameter is selected to be tested. Food safety standards also require testing on Salmonella, Listeria monocytogenes, Escherichia coli.

Enterobacteriaceae and number of yeast and molds could also be tested to provide full microbiological profile during shelf life.

The results and discussion section is appropriate. Some needed explanations are noted in the text.

Conclusion section is comprehensive and concludes the idea presented in this reserach.

Some minor corrections are noted in manuscripts’ pdf file.

Reviewer recommendation: Major revision.

Author Response

Dear Reviewer 3,

Thank you so much for you revision and suggestions. We have improve the manuscript following the suggestions raised by reviewers. The whole manuscript has been revised and some sentences have been rephrased for a better clarity.

Below are the replies to your specific comments.

  • The Materials and Methods section describes all conducted testing. Regarding microbiological testing, there is question why only one parameter is selected to be tested. Food safety standards also require testing on Salmonella, Listeria monocytogenes, Escherichia coli.
  • Enterobacteriaceae and number of yeast and moulds could also be tested to provide full microbiological profile during shelf life.

Regarding microbiological analysis, moulds and yeasts were also tested, but no significant colonies growth was detected in any sample. For this reason, it was decided not to report the data, and therefore the determination methodology used, in the manuscript. In addition, regarding foodborne pathogens, in a previous study (Imeneo, V., Romeo, R., De Bruno, A., Piscopo, A. (2022).  Green-sustainable extraction techniques for the recovery of antioxidant compounds from “citrus Limon” by-products. Journal of Environmental Science and Health, Part B. https://doi.org/10.1080/03601234.2022.2046993), of which this is a continuation, the antimicrobial activity of the lemon by-product used in the formulation of the pectin-based coating was tested against certain foodborne pathogens (such as Salmonella, L. monocytogenes and E. Coli). In this regard, the extract showed antimicrobial activity against all three pathogens tested.

  • The results and discussion section is appropriate. Some needed explanations are noted in the text.

We thank you for your suggestions and we have modified the text as suggested.

  • Some minor corrections are noted in manuscripts’ pdf file.

Regarding the increase of TCC in W and PC fresh cut carrots (lines 436-437), the explanation is given later in the text (lines 467-471).

Round 2

Reviewer 2 Report

The corrections are well done. Good luck to you dear researchers

Reviewer 3 Report

Changes in the manuscript have impoved its' quality.